# Outcomes of Percutaneous Coronary Interventions for Long Diffuse Coronary Artery Disease with Extremely Small Diameter

**DOI:** 10.3390/jcm12041285

**Published:** 2023-02-06

**Authors:** Chien-Te Ho, Fu-Chih Hsiao, Ying-Chang Tung, Sharon T. Cordero, Dominador V. del Castillo, Hsin-Fu Lee, Shing-Hsien Chou, Chia-Pin Lin, Kun-Chi Yen, Lung-An Hsu, Chi-Jen Chang

**Affiliations:** Cardiovascular Division, Chang Gung Memorial Hospital, Chang Gung University College of Medicine, Taoyuan 33353, Taiwan

**Keywords:** long diffuse lesions, extremely small vessel, DES, outcome

## Abstract

Background. The optimal percutaneous coronary intervention (PCI) strategy and clinical outcomes of long lesions with an extremely small residual lumen remain unclear. This study aimed to assess the efficacy of a modified stenting strategy for diffuse coronary artery disease (CAD) with an extremely small distal residual lumen. Methods. 736 Patients who received PCI using second-generation drug-eluting stents (DES) ≥38 mm long were retrospectively included and categorized into an extremely small distal vessel (ESDV) group (≤2.0 mm) and a non-ESDV group (>2.0 mm) according to the maximal luminal diameter of the distal vessel (dsD_Max_). A modified stenting technique was applied by landing an oversized DES in the distal segment with the largest luminal diameter and maintaining the distal stent edge partially expanded. Results. The mean dsD_Max_ and stent lengths were 1.7 ± 0.3 mm and 62.6 ± 18.1 mm in the ESDV group and 2.7 ± 0.5 mm and 59.1 ± 16.0 mm in non-ESDV groups, respectively. The acute procedural success rate was high in both the ESDV and non-ESDV groups (95.8% and 96.5%, *p* = 0.70) with rare distal dissection (0.3% and 0.5%, *p* = 1.00). The target vessel failure (TVF) rate was 16.3% in the ESDV group and 12.1% in the non-ESDV group at a median follow-up of 65 months without significant differences after propensity score matching. Conclusions. PCI using contemporary DES with this modified stenting technique is effective and safe for diffuse CAD with extremely small distal vessels.

## 1. Introduction

Substantial clinical evidence supports the efficacy and safety of percutaneous coronary intervention (PCI) using the current generation drug-eluting stents (DESs) for treating long diffuse coronary lesions [1,2,3,4,5]. However, the optimal PCI strategy and clinical outcomes remain uncertain for long lesions extending to the distal vessel segment with a diffuse small residual lumen, particularly those with an extremely small luminal diameter of ≤2.0 mm. Diffuse small vessels make it challenging to achieve optimal procedural results using standard stenting techniques. PCI using even the smallest-sized device (2.0–2.25 mm) of currently available DESs is oversized for lesions with a luminal diameter ≤ 2.0 mm and carries a risk of distal edge dissection. Additionally, limitations in implanting an oversized stent may hamper complete lesion coverage and lead to a higher incidence of target vessel revascularization (TVR) [6,7,8]. A modified stenting strategy of implanting an oversized DES by landing the distal edge in the segment with the largest diameter of the distal vessel and leaving the distal edge partially expanded may achieve a higher lesion coverage with a lower risk of distal edge dissection in such lesions.

In this study, we investigated the acute procedural and long-term clinical outcomes in patients with long diffuse coronary artery disease (CAD) and maximal diameter of the distal segment (dsD_Max_) measuring ≤2.0 mm treated using the modified stenting technique and current-generation DESs. We sought to evaluate patient outcomes in this at-risk population and compared them with those with a distal lumen of >2.0 mm.

## 2. Methods

### 2.1. Study Population and Covariates

Consecutive patients who underwent PCI at Chang Gung Memorial Hospital, LinKou, Taiwan, between January 2010 and December 2017 were screened retrospectively for eligibility. We identified cases from the medical records with at least one long coronary lesion treated with one or multiple overlapping second-generation DESs with a total stent length of ≥38 mm. We excluded patients presenting with ST-segment elevation myocardial infarction (STEMI) or cardiogenic shock. The included patients with dsD_Max_ ≤ 2.0 mm based on the quantitative coronary analysis were categorized into the extremely small distal vessel (ESDV) group, and the remaining patients were assigned to the non-ESDV group (dsD_Max_ > 2.0 mm). Patients with simultaneous long lesions of dsD_Max_ ≤ 2.0 and dsD_Max_ > 2.0 mm in different vessels were categorized into the ESDV group. The baseline characteristics, risk factors for atherosclerotic cardiovascular disease, comorbidities, and medications of the included patients were obtained from the electronic medical records.

The study was approved by the Institutional Review Board of Chang Gung Memorial Hospital. All the patients underwent standard medical management. The requirement for written consent from patients was waived owing to the retrospective nature of this study.

### 2.2. PCI Procedure

All interventions were performed according to standard techniques, except for the modified stent deployment technique. For lesions with a small distal luminal diameter ≤ 2.0 mm, a modified stenting technique was used (Figure 1). To accommodate the oversized stent and achieve a higher lesion coverage, the segment with the largest luminal diameter of the distal vessel was considered the distal landing zone. During deployment, the stent was slowly inflated with each atmosphere every 2–3 s. The initial stent deployment pressure was approximately 5–7 atm to maintain partial stent expansion. Additional post-dilatation was performed using an appropriate-sized non-compliant (NC) balloon for different segments. To maintain the distal stent edge under-expanded, the distal 1–2 mm of the stent was spared for post-dilation. This modified technique was the standard of care for this specific type of lesion and has been uniformly performed by all operators in our catheterization laboratory since the era of first-generation DES. The second-generation DES used included Xience (Abbott Vascular, Santa Clara, CA, USA), Synergy (Boston Scientific, Natick, MA, USA), Ultimaster (TERUMO, Tokyo, Japan), and Resolute (Medtronic, Santa Rosa, CA, USA). Quantitative coronary analysis was performed retrospectively using Centricity Universal Viewer (GE Healthcare, version 6.0, Chicago, IL, USA) by two indepe ndent cardiologists who were blinded to clinical information. The measurements were performed during diastole after intracoronary nitroglycerin administration using a guiding catheter to calibrate the magnification. 

All patients were pretreated with standard-dose dual antiplatelet therapy before and after the PCI procedure, according to the clinical condition and judgment of the primary care physician.

### 2.3. Definitions and Clinical Follow-Up

Procedural success was defined as the attainment of residual diameter stenosis of ≤30% and thrombolysis in myocardial infarction (TIMI) flow grade 3 without angiographic dissection. The primary clinical endpoint was target vessel failure (TVF), defined as a composite of cardiac death, target vessel myocardial infarction (TVMI), and clinically driven-target vessel revascularization (TVR). Death was considered cardiac in origin unless obvious noncardiac causes could be identified. TVMI was defined as myocardial infarction that could be related to the target vessel but not to another vessel. TVR was defined as any repeat percutaneous intervention or coronary artery bypass graft (CABG) of any segment of the target vessel. TVR was considered clinically driven if revascularization was performed in patients with ischemic signs confirmed using noninvasive tests or functional assessment, irrespective of the ischemic symptoms. Definite or probable stent thrombosis was defined according to the Academic Research Consortium criteria [9]. For the ESDV patients with simultaneous lesions with dsD_Max_ ≤ 2.0 and >2.0 mm in different vessels, only events related to the vessel with dsD_Max_ ≤ 2.0 mm were counted. All patients were clinically followed up via clinic visits or telephone interviews.

### 2.4. Statistical Analysis

The Chi-square test and Fisher’s exact test, if necessary, were used to compare the categorical variables. The two-sample *t*-test was used to test continuous variables, and the results were expressed as mean ± standard deviation (SD). Propensity score matching was performed at a ratio of 1:1 between the groups. The following variables were used in this analytical model: age, sex, smoking status, diabetes mellitus, hypertension, hyperlipidemia, chronic kidney disease, peripheral artery disease, left ventricular ejection fraction < 40%, clinical presentation of MI, history of MI, history of stroke, number of diseased vessels, culprit vessel, left main disease, chronic total occlusion (CTO), moderate-to-severe calcification, duration of dual-antiplatelet therapy, and stent length. The match tolerance was set to 0.01. The survival curve was plotted using the Kaplan–Meier method, and significance was examined using the log-rank test. Univariate Cox proportional hazard analysis was used to evaluate potential predictors of TVF. All variables (*p* < 0.05) in the univariate Cox regression analysis were then entered into the multivariate Cox proportional hazard model to determine the independent predictors of long-term TVF. *p* < 0.05 for a two-sided test was considered statistically significant. All statistical analyses were performed using SPSS version 20.0 (IBM, Inc., Chicago, IL, USA). 

## 3. Results

### 3.1. Clinical and Angiographic Characteristics

Among the screened patients, 736 were eligible for analysis. Of these, 307 (42%) patients were categorized into the ESDV group and the remaining 429 (58%) into the non-ESDV group. The baseline and angiographic characteristics of the study population are summarized in Table 1. Patients in the ESDV group were more likely to be female, had a higher prevalence of chronic kidney disease (33.9% vs. 25.9%; *p* = 0.019), and a lower prevalence of smoking (26.4% vs. 33.6%; *p* = 0.037) compared with the non-ESDV group. The mean dsD_Max_ was 1.7 ± 0.3 mm and 2.7 ± 0.5 mm in ESDV and non-ESDV groups, respectively (*p* < 0.005). The ESDV group had more CTO lesions than the non-ESDV group (46.3% vs. 19.6%, *p* < 0.001).

### 3.2. Procedural Details and Outcomes

Procedural details and outcomes are summarized in Table 2. The mean initial pressure of deployment for the most distal stent was 6.6 ± 1.8 atm in the ESDV group and 8.9 ± 3.1 atm in the non-ESDV group (*p* < 0.0001). High-pressure post-dilatation was performed in 99.3% of patients in both groups. There was no significant difference in the number of stents (1.9 ± 0.5 vs. 1.9 ± 1.1; *p* = 0.46). However, patients in the ESDV group had a longer stent length (62.6 ± 18.1 mm vs. 59.1 ± 16.0 mm; *p* = 0.007) and a smaller mean stent diameter (2.8 ± 0.3 mm vs. 3.0 ± 0.4 mm; *p* < 0.001) than those in the non-ESDV group.

The acute procedural success rates were similar between the ESDV and non-ESDV groups (95.8% and 96.5%, respectively; *p* = 0.7). Angiographically detectable distal edge dissection was observed in one (0.3%) patient in the ESDV group and two (0.5%) patients in the non-ESDV group and was successfully rescued by additional stenting in all cases.

### 3.3. Long-Term Clinical Outcomes

The median follow-up was 65.4 months (interquartile range: 50.9–89.8 months) in the ESDV group and 64.5 months (interquartile range: 51.3–88.8 months) in the non-ESDV group (*p* = 0.8). The rate of complete follow-up was 94.1% and 96.5% (*p* = 0.15) in the ESDV and the non-ESDV groups, respectively. As shown in Table 3, the TVF rates were 16.3% (50/307) and 12.1% (52/429) in the ESDV and non-ESDV groups, respectively. The Kaplan–Meier curve of TVF is illustrated in Figure 2. The rates of cardiac death, TVMI, and CDTVR in the ESDV and non-ESDV groups were 3.9% and 3.5%, 4.9% and 3.0%, and 12.7% and 9.1%, respectively. Probable or definite stent thrombosis was observed in three (1%) patients in the ESDV group and one (0.2%) in the non-ESDV group.

After propensity score matching, 237 paired patients were well-matched in terms of baseline characteristics between the two groups. After matching, the TVF rate did not differ between the ESDV and non-ESDV groups (13.5% vs. 13.1; hazard ratio, 1.09; 95% confidence interval, 0.67–1.79, *p* = 0.73) along with cardiac death, all-cause mortality, TVMI, CDTVR, stroke, and stent thrombosis (Table 3). Figure 3 shows the comparisons of the cumulative incidence rates of TVF and its components between the two study groups after matching. There were no significant differences in the rates of TVF, CV death, TVMI, and CDTVR.

### 3.4. Predictors of TVF

Cox regression analysis was performed to determine whether dsD_Max_ ≤ 2 mm was a predictor of TVF. Univariate analysis revealed female sex, diabetes mellitus, hypertension, chronic kidney disease, peripheral artery disease, history of MI or stroke, LM disease, and moderate or severe coronary calcification as predictors of TVF. However, dsD_Max_ ≤ 2 mm was not identified as a predictor. The multivariate Cox proportional hazards model revealed that diabetes mellitus and chronic kidney disease were independent predictors of TVF (Table 4). Considering the potential feature of the negative remodeling in vessels distal to the CTO segments [10], resulting in inappropriate patient categorization according to dsD_Max_, Cox regression analysis was performed with the exclusion of patients with CTO lesions, revealing that dsD_Max_ ≤ 2 mm was still not a predictor for TVF.

## 4. Discussion

In the present study, we demonstrated that PCI using the current-generation DES and a modified stenting technique was effective and safe in treating long diffuse coronary lesions with extremely small dsD_Max_ ≤ 2.0 mm, with acute procedural and long-term clinical outcomes comparable to those of long diffuse lesions with dsD_Max_ > 2.0 mm. Given the extremely small distal lumen, long lesion length, and complex baseline clinical characteristics, the acute procedural success was 95.8%, and the TVF rate was 16.3% during a median follow-up period of 65 months.

Substantial evidence supports the effectiveness and safety of PCI using current-generation DES for long diffuse coronary lesions. Clinical studies have shown that treating long diffuse lesions with multiple DESs > 60 mm long resulted in favorable outcomes, with a TVF rate of 10–13.6% [1,2,3,4,5,11,12], similar to those of both groups in our study. Current-generation DESs have also been demonstrated to be feasible for small-vessel diseases. In a substudy of the DUrable polymer-based sTent CHallenge of Promus ElemEnt versus ReSolute integrity: DUTCH PEERS (TWENTE II) trial, the reported rate of target lesion failure for small target vessels measuring <2.5 mm treated using second-generation DES was 9.5% at the 2-year follow-up [13]. Moreover, in the prespecified substudy of the Comparison of Biodegradable Polymer and Durable Polymer Drug-eluting Stents in an All-Comers Population (BIO-RESORT) trial, treating small vessels measuring <2.5 mm (mean reference vessel diameter: 2.11 mm) using three different second-generation DESs was reported to result in a TVF rate of 7.0–9.5% at the 3-year follow-up [14]. Recently, the smallest ZES (2.0 mm) has been made available. In a study with a relatively small number of patients, using this stent for short lesions with a mean reference vessel diameter (RVD) of 1.91 mm resulted in a TLF rate of 5% at 1 year [15]. In the ESDV group in our study, the long diffuse disease was accompanied by an extremely small dsD_Max_, smaller than those of the lesions included in the studies mentioned above. Challenges exist in the implantation of stents for this group of lesions. For the diffuse small residual lumen measuring ≤2.0 mm, PCI using even the smallest DES is oversized, which may increase the risk of distal edge dissection [15] and hamper complete lesion coverage. To achieve higher lesion coverage with a lower risk of edge dissection, we modified the stenting strategy by landing the distal edge of an oversized stent at the segment with the largest diameter of the distal vessel and maintaining the distal stent edge partially expanded. We demonstrated that this modified strategy was associated with a low rate of distal-edge dissection (0.3%). There are concerns that leaving the distal stent edge under-expanded may increase the risk of stent thrombosis and in-stent restenosis [16]. Given the mean DAPT duration of 10 months in this study, a stent thrombosis rate of 1.0% was observed in the ESDV group. Additionally, accompanied by the long stent length and the limitation in complete lesion coverage, this modified strategy resulted in a clinically driven TVR rate of 12.7% over a median follow-up period of >5 years, similar to that of the non-ESDV group. These results indicate that intentionally leaving the distal stent edge under-expanded, with acceptable stent apposition for the landing zone of vessels with extremely small diameters, is safe and effective for this group of lesions.

In this study, high-pressure post-dilatation was performed in 99.3% of all lesions using an appropriate-sized NC balloon according to the reference vessel diameter of each stented segment at a mean pressure of 19.2 ± 4.3 atm. For distal stented segments, a 2.5-mm NC balloon was routinely used for post-dilation to achieve adequate stent expansion and a larger stent area, except for the distal stent edge. This post-dilation strategy may have contributed to the favorable TVR and stent thrombosis rates in the ESDV group.

CABG is an alternative therapeutic option for patients with long-diffuse ESDV [17]. In this clinical setting, diffuse disease leads to higher SYNTAX scores, with PCI tending to be associated with worse outcomes than CABG. However, CABG for a target vessel with a small diameter has been shown to be associated with worse outcomes [18,19]. In patients who underwent CABG for the mid-left anterior descending artery, an early study demonstrated a markedly high in-hospital mortality rate of 15.8% for vessels with a diameter of 1.0-mm and 4.6% for vessels with a diameter of 1.5–2.0 mm compared with a rate of 1.5% for vessels with a diameter of 2.5–3.5 mm [19]. A more recent randomized study demonstrated decreased 1-year graft patency in patients with small vessel disease who underwent CABG for a non-left anterior descending artery with either arterial or venous conduit, especially in cases of graft–target size mismatch [20].

Drug-coated balloon (DCB) is another potential treatment for small and very small vessels, and it has been shown to be non-inferior to DES in recent years. However, the lesions treated with DCB in these clinical trials were short (ranging from 10.5–23 mm in length) with a relatively larger vessel diameter (ranging from 2.11–2.75 mm in size) [21,22,23,24], representing a different lesion set from lesions treated in the present study. In Assess the Efficacy and Safety of RESTORE Paclitaxel Eluting Balloon Versus RESOLUTE Zotarolimus Eluting Stent for the Treatment of Small Coronary Vessel Disease (RESTORE SVD China) study, a specified very small vessel cohort treated with DCB was enrolled in a nested registry. The vessel diameter of those lesions (1.86 ± 0.28 mm) was similar to the distal luminal diameter of the ESDV group in our study, although the lengths of lesions were apparently shorter (length: 12.2 ± 5.6 mm) [24]. Treatment with DCB in this cohort revealed an apparently higher 9-month in-segment stenosis rate of 38.4% than the cohort with vessel diameter larger than 2.25 mm, underscoring that treating lesions with very small vessel sizes using DCB may result in worse clinical outcomes. Further clinical trials to compare these treatment options are required to confirm the revascularization modality of choice for this specific group of lesions.

## 5. Limitations

Our study had several limitations. First, this was a single-center retrospective study; therefore, it is inherently prone to bias. Because that patients with the long diffuse disease and an extremely small distal lumen represented a small proportion of patients treated with PCI, the patient number in this study was also limited. Seconds, patients estimated by operators to have a high procedural risk may have avoided PCI. The findings of this study cannot be extrapolated to all long ESDV lesions. Third, this study lacked a control group with ESDV treated using the standard stenting technique, which is not feasible in terms of the risk of distal edge dissection. Therefore, we could not conclude any improvement in the outcomes with the modified stenting technique. Fourth, intravascular imaging was performed in a relatively small proportion of patients. Angiography alone was not capable of evaluating atherosclerotic burden and distinguishing between true small vessels and diffuse diseases with small residual diameters. Last, at the segment distal to the stent of the target vessel, significant stenosis could exist because of diffuse atherosclerosis. Stenting was hardly possible for this segment. Even if adjunctive balloon dilation was performed, residual ischemia could remain after the procedure. Further prospective multicenter trials enrolling larger patient numbers with the guidance of intravascular imaging and functional assessment are mandatory to validate the efficacy and safety of this modified stenting technique in treating the long diffuse lesion with an extremely small luminal diameter.

## 6. Conclusions

The present study demonstrated that PCI using a contemporary DES and a modified stenting technique resulted in high acute procedural success and favorable long-term clinical outcomes in long diffuse coronary lesions with an extremely small residual lumen. Larger prospective studies are required to corroborate the findings of this retrospective analysis.

## Figures and Tables

**Figure 1 jcm-12-01285-f001:**
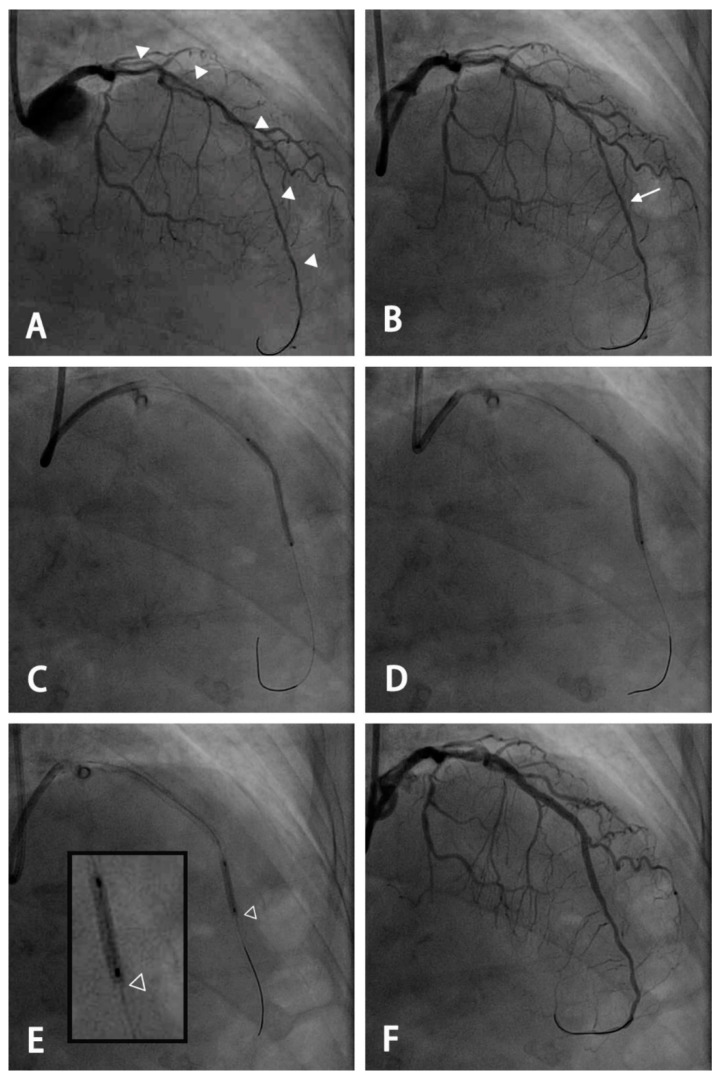
An example of the use of a modified stenting technique for a long diffuse lesion with an extremely small distal vessel. (**A**) A long diffuse lesion of the left anterior descending artery extending from the proximal to the distal segment with a small residual lumen of <2.0 mm (arrowhead) using the 7 French EBU catheter as a reference. (**B**) Angiography after pre-dilatation and intracoronary administration of nitroglycerin. A planned distal landing zone was at the segment with the largest luminal diameter (arrow). (**C**) An oversized 2.5 mm drug-eluting stent was deployed and inflated at 5 atmospheres, with the stent remaining partially expanded. (**D**) Further dilation using the stent balloon was performed at the nominal pressure after pulling the balloon back by 1–2 mm to keep the distal stent edge under-expanded. (**E**) Post-dilatation was performed using a 2.5-mm non-compliant balloon with the most distal 2 mm spared (hollow arrowhead: the distal edge of the stent). (**F**) Final angiogram after stenting for the proximal segment and post-dilatation for each segment using a non-compliant balloon of appropriate size.

**Figure 2 jcm-12-01285-f002:**
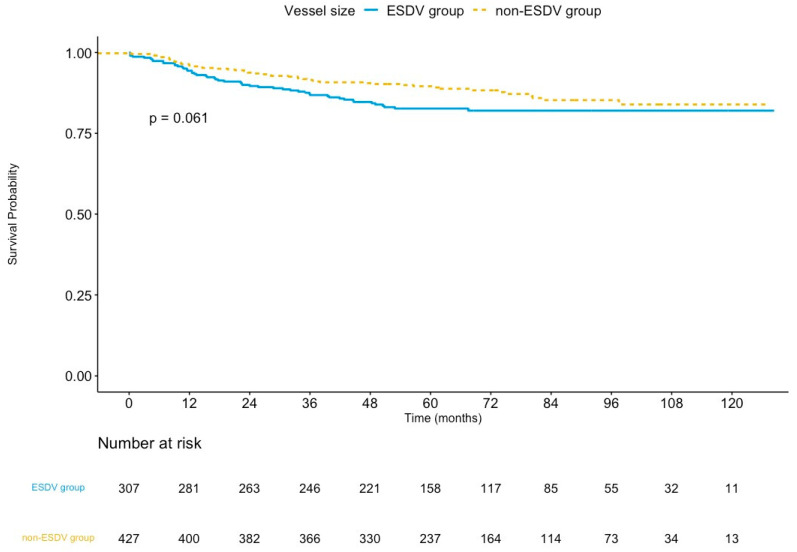
Kaplan–Meier survival curves of target vessel failure comparing the ESDV to the non-ESDV group in general study populations. ESDV indicates extensive small distal vessels.

**Figure 3 jcm-12-01285-f003:**
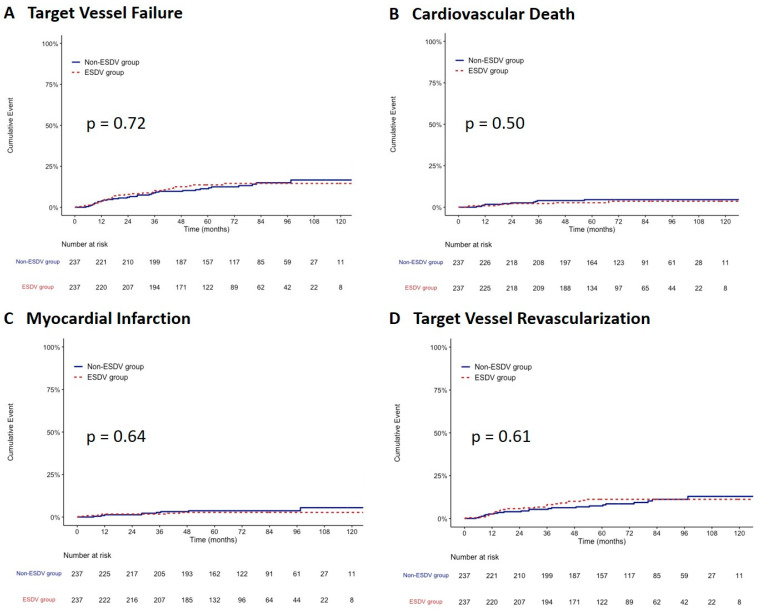
Kaplan–Meier survival curves comparing the ESDV to the non-ESDV group in matched pairs (237 pairs). (**A**) Target vessel failure, (**B**) Cardiovascular death, (**C**) Myocardial infarction, and (**D**) Target vessel revascularization. ESDV indicates extensive small distal vessels.

**Table 1 jcm-12-01285-t001:** Baseline Clinical and Angiographic Characteristics.

	ESDV Group N = 307	Non-ESDV Group N = 429	*p* Value
Age (years)	65.1 ± 11.8	64.5 ± 10.8	0.52
Male	232(75.6)	358(83.0)	0.013
Presentation of myocardial infarction	79(25.7)	120(28.0)	0.50
Diabetes mellitus	163(53.1)	209(48.7)	0.24
Hypertension	235(76.5)	332(77.4)	0.79
Hyperlipidemia	268(87.3)	355(82.8)	0.09
Smoking	81(26.4)	144(33.6)	0.037
Chronic kidney disease	104(33.9)	111(25.9)	0.019
left ventricular ejection fraction < 40%	31(10.1)	40(9.3)	0.73
History of myocardial infarction	25(8.1)	52(12.1)	0.08
History of stroke	12(3.9)	24(5.6)	0.30
Peripheral artery disease	27(8.8)	45(10.5)	0.45
Number of vessel disease			0.56
1-vessel disease	48(15.6)	78(18.2)	
2-vessel disease	109(35.5)	156(36.4)	
3-vessel disease	150(48.9)	195(45.5)	
Culprit vessel			0.24
Left anterior descending artery	187(60.9)	241(56.2)	
Left circumflex artery	39(12.7)	50(11.7)	
Right coronary artery	81(26.4)	138(32.2)	
Left main coronary disease	45(14.7)	69(16.1)	0.60
Chronic total occlusion	142(46.3)	84(19.6)	<0.001
Distal vessel maximal diameter (mm)	1.7 ± 0.3	2.7 ± 0.5	<0.005
Moderate to severe calcification	113(36.8)	165(38.5)	0.65
Duration of DAPT (months)	10.3 ± 3.0	10.4 ± 3.0	0.70

Values are expressed as mean ± SD or n (%). DAPT, dual antiplatelet therapy; ESDV, extensive small distal vessel; SD, standard deviation.

**Table 2 jcm-12-01285-t002:** Procedural Characteristics.

	ESDV Group N = 307	Non-ESDV Group N = 429	*p* Value
Stent length (mm)	62.6 ± 18.1	59.1 ± 16.0	0.007
Number of stents	1.9 ± 0.5	1.9 ± 1.1	0.46
Mean stent diameter (mm)	2.8 ± 0.3	3.0 ± 0.4	<0.001
Stent deployment pressure (atm)	6.6 ± 1.8	8.9 ± 3.1	<0.0001
Post-dilatation	305(99.3)	426(99.3)	1.00
Post-dilatation pressure (atm)	18.6 ± 4.0	19.6 ± 4.4	0.10
Image guidance	59(19.2)	70(16.3)	0.33
Residual diameter stenosis (%)	14.4 ± 9.6	14.9 ± 9.5	0.97
Patients with residual diameter stenosis >30%	10(3.0)	11(2.7)	1.00
Post-PCI TIMI flow grade < 3	2(0.7)	2(0.5)	1.00
Distal edge dissection	1(0.3)	2(0.5)	1.00
Acute stent thrombosis	0(0)	0(0)	1.00
Procedural success	294(95.8)	414(96.5)	0.70

Values are expressed as mean ± SD or n (%). ESDV, extensive small distal vessel; PCI, percutaneous coronary intervention; SD, standard deviation; TIMI, thrombolysis in myocardial infarction.

**Table 3 jcm-12-01285-t003:** Long-Term Clinical Outcomes.

	Overall Population N = 736				Propensity Score Matching-Matched Patients N = 474			
	ESDV Group N = 307	Non-ESDV Group N = 429	Hazard Ratio (95% CI)	*p* Value	ESDV Group N =237	Non-ESDV Group N =237	Hazard Ratio (95% CI)	*p* Value
Target vessel failure	50(16.3)	52(12.1)	1.38(0.93–2.04)	0.10	32(13.5)	31(13.1)	1.09(0.67–1.79)	0.73
Cardiac death	12(3.9)	15(3.5)	1.12(0.52–2.39)	0.77	7(3.0)	10(4.2)	0.72(0.27–1.89)	0.50
TVMI	15(4.9)	13(3.0)	1.59(0.76–3.35)	0.22	7(3.0)	9(3.8)	0.79(0.30–2.13)	0.64
Clinical driven TVR	39(12.7)	39(9.1)	1.45(0.93–2.26)	0.10	24(10.1)	22(9.3)	1.16(0.65–2.07)	0.61
death	49(16.0)	66(15.4)		0.83	35(14.8)	47(19.8)		0.15
stroke	5(1.6)	8(1.9)		0.81	2(0.8)	4(1.7)		0.69
Stent thrombosis	3(1.0)	1(0.2)		0.31	1(0.4)	0(0)		1.00

Values are expressed as n (%). TVR, driven target vessel revascularization; CI, confidence interval; ESDV, extensive small distal vessel; TVMI, target vessel myocardial infarction.

**Table 4 jcm-12-01285-t004:** Cox Proportional Hazard Analysis of Predictors of Target Vessel Failure.

	Univariate Analysis		Multivariate Analysis	
	Hazard Ratio (95% CI)	*p* Value	Hazard Ratio (95% CI)	*p* Value
Age	1.005(0.987–1.023)	0.59		
Male	0.546(0.355–0.840)	0.006	0.732(0.467–1.145)	0.17
Presentation of myocardial infarction	1.056(0.680–1.639)	0.81		
Diabetes mellitus	2.410(1.586–3.664)	<0.001	1.679(1.077–2.620)	0.022
Hypertension	1.997(1.135–3.511)	0.016	1.321(0.730–2.392)	0.36
Hyperlipidemia	1.243(0.680–2.270)	0.48		
Smoking	0.859(0.559–1.321)	0.49		
Chronic kidney disease	2.753(1.862–4.068)	<0.001	1.826(1.175–2.836)	0.007
Left ventricular ejection fraction < 40%	2.062(1.191–3.571)	0.01	1.233(0.685–2.218)	0.49
Old myocardial infarction	1.772(1.038–3.025)	0.036	1.504(0.864–2.618)	0.15
Old stroke	2.216(1.117–4.397)	0.023	1.358(0.665–2.773)	0.40
peripheral artery disease	2.986(1.809–4.927)	<0.001	1.571(0.889–2.774)	0.12
Number of diseased vessels	1.141(0.612–2.127) 1.402(0.779–2.524)	0.68 0.26		
Culprit vessel	1.340(0.974–1.843) 0.671(0.401–1.123)	0.072 0.13		
Left main disease	1.657(1.041–2.636)	0.033	1.336(0.828–2.154)	0.26
CTO	1.110(0.736–1.672)	0.62		
Duration of DAPT	0.976(0.912–1.043)	0.47		
Moderate to severe calcification	1.865(1.264–2.751)	0.002	1.295(0.851–1.969)	0.23
stent length, mm	1.007(0.996–1.017)	0.23		
stent number	1.051(0.866–1.275)	0.62		
stent diameter (mean)	0.802(0.449–1.435)	0.46		
dsD_Max_ ≤ 2 mm	1.381(0.937–2.037)	0.10		

CI, confidence interval; CTO, chronic total occlusion; DAPT, dual-antiplatelet therapy; dsD_Max_, maximal luminal diameter of the distal segment; ESDV, extensive small distal vessel; SD, standard deviation.

## Data Availability

The datasets used and/or analyzed during the current study are available from the corresponding author upon request.

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
