# Peer review of "Outcomes of Percutaneous Coronary Interventions for Long Diffuse Coronary Artery Disease with Extremely Small Diameter"

_jcm, 2023, doi:10.3390/jcm12041285_

Round 1

Reviewer 1 Report

1.The data used in the study are now five years old, which is relatively old, and the article will be more abundant and more detailed if it can be supplemented with data from recent years.

2.Some references are older.

3.The language expression of the article requires a certain amount of embellishment and polishing.

Reviewer 2 Report

The article "Outcomes of Percutaneous Coronary Interventions for Long Diffuse Coronary Artery Disease with Extremely Small Diameter" concerns an interesting modified stenting technique, applied by landing an oversized DES in the distal segment with the largest luminal diameter and maintaining the distal stent edge partially expanded. Overall, the manuscript is well-written and organized. However, I have a few comments:

-          Was the above technique verified angiographically in the enrolled patients?

-          Please add a p-value for the obtained values in summary.

-          How many patients underwent intracoronary imaging?

-          Is the DAPT of 10 months an average value? Was it a general recommendation for all patients?

-          The authors do not mention other treatment options for small and very small vessels, e.g. DCB.

Reviewer 3 Report

The manuscript is well written and findings are of potential interest. However, I have some concerns with the study design and presentation of results. In particular, the sample size of the study is limited to derive meaningful conclusions.

1) As above reported, I have some concerns with the reliability of results due to the limited number of patients included. In my opinion even surrogate endpoints needs larger cohorts. I suggest to include, if possible, a larger number of patients in order to achieve adequate statistical power.

2) The little usage of coronary imaging, although already stated in the limitation section by the author, represent a huge limitation in the hypothetical evaluation of a new or modified stent technique.  
